# Derivation and external validation of a risk score for predicting HIV-associated tuberculosis to support case finding and preventive therapy scale-up: A cohort study

Andrew F. Auld[1]*, Andrew D. Kerkhoff[2], Yasmeen Hanifa[3], Robin Wood[4], Salome Charalambous[5], Yuliang Liu[1], Tefera Agizew[6], Anikie Mathoma[6,7], Rosanna Boyd[6], Anand Date[1], Ray W. Shiraishi[1], George Bicego[1], Unami Mathebula-Modongo[6], Heather Alexander[1], Christopher Serumola[6], Goabaone Rankgoane-Pono[8], Pontsho Pono[8], Alyssa Finlay[6], James C. Shepherd[6,9], Tedd V. Ellerbrock[1], Alison D. Grant[3,10ↄ], Katherine Fielding[3ↄ]

1 Division of Global HIV & TB, Centers for Disease Control and Prevention, Atlanta, Georgia, United States of America, 2 Division of HIV, Infectious Diseases and Global Medicine at Zuckerberg San Francisco General Hospital and Trauma Center, Department of Medicine, University of California San Francisco, San Francisco, California, United States of America, 3 TB Centre, London School of Hygiene & Tropical Medicine, London, United Kingdom, 4 The Desmond Tutu HIV Centre, Institute of Infectious Disease and Molecular Medicine, Cape Town, South Africa, 5 Aurum Institute, Johannesburg, South Africa, 6 Division of TB Elimination, Centers for Disease Control and Prevention, Gaborone, Botswana, 7 University of Botswana, Gaborone, Botswana, 8 Ministry of Health and Wellness, Gaborone, Botswana, 9 Yale University School of Medicine, New Haven, Connecticut, United States of America, 10 Africa Health Research Institute, School of Laboratory Medicine and Medical Sciences, College of Health Sciences, University of KwaZulu-Natal, Durban, South Africa

ↄ These authors contributed equally to this work.
* aauld@cdc.gov

**Data Availability Statement:** The authors confirm that, for IRB-approved reasons, some access

## Abstract

### Background

Among people living with HIV (PLHIV), more flexible and sensitive tuberculosis (TB) screening tools capable of detecting both symptomatic and subclinical active TB are needed to (1) reduce morbidity and mortality from undiagnosed TB; (2) facilitate scale-up of tuberculosis preventive therapy (TPT) while reducing inappropriate prescription of TPT to PLHIV with subclinical active TB; and (3) allow for differentiated HIV–TB care.

### Methods and findings

We used Botswana XPRES trial data for adult HIV clinic enrollees collected during 2012 to 2015 to develop a parsimonious multivariable prognostic model for active prevalent TB using both logistic regression and random forest machine learning approaches. A clinical score was derived by rescaling final model coefficients. The clinical score was developed using southern Botswana XPRES data and its accuracy validated internally, using northern Botswana data, and externally using 3 diverse cohorts of antiretroviral therapy (ART)-naive and ART-experienced PLHIV enrolled in XPHACTOR, TB Fast Track (TBFT), and Gugulethu studies from South Africa (SA). Predictive accuracy of the clinical score was compared

restrictions apply to the data underlying the findings. Although the patient-level data do not include patient names, this IRB decision is in the interest of ensuring patient confidentiality. An individual may email the CDC Division of Global HIV & TB science office (gapmts@cdc.gov) to request the data.

**Funding:** This research has been supported by the President's Emergency Plan for AIDS Relief (PEPFAR) through the US Centers for Disease Control and Prevention. The funder had no role in the study design, data collection and analysis, decision to publish, or preparation of the manuscript.

**Competing interests:** I have read the journal's policy and the authors of this manuscript have the following competing interests: AG has received research grants (not related to this work) from the Medical Research Council, Economic and Social Research Council, Bill and Melinda Gates Foundation, National Institute of Allergy and Infectious Diseases, Wellcome Trust, Research England, and USAID.

**Abbreviations:** AIC, Akaike information criteria; ART, antiretroviral therapy; AUROC, area under the receiver operating characteristic; BIC, Bayesian information criteria; BMI, body mass index; CDC, Centers for Disease Control and Prevention; COVID-19, Coronavirus Disease 2019; CRP, C-reactive protein; CRT, cluster randomized trial; CXR, chest radiography; EMR, electronic medical record; HRDC, Health Research and Development Committee; IRB, Institutional Review Board; LMIC, low- and middle-income country; MFP, multivariable fractional polynomial; NNS, number needed to screen; NPV, negative predictive value; PEPFAR, US President's Emergency Plan for AIDS Relief; PLHIV, people living with HIV; POC, point-of-care; PPV, positive predictive value; SA, South Africa; SOC, standard of care; SSA, sub-Saharan Africa; TB, tuberculosis; TBFT, TB Fast Track; TPT, tuberculosis preventive therapy; TRIPOD, Transparent Reporting of a multivariable prediction model for Individual Prognosis or Diagnosis; WHO, World Health Organization.

with the World Health Organization (WHO) 4-symptom TB screen. Among 5,418 XPRES enrollees, 2,771 were included in the derivation dataset; 67% were female, median age was 34 years, median CD4 was 240 cells/μL, 189 (7%) had undiagnosed prevalent TB, and characteristics were similar between internal derivation and validation datasets. Among XPHACTOR, TBFT, and Gugulethu cohorts, median CD4 was 400, 73, and 167 cells/μL, and prevalence of TB was 5%, 10%, and 18%, respectively. Factors predictive of TB in the derivation dataset and selected for the clinical score included male sex (1 point), $\geq$1 WHO TB symptom (7 points), smoking history (1 point), temperature >37.5˚C (6 points), body mass index (BMI) <18.5kg/m$^2$ (2 points), and severe anemia (hemoglobin <8g/dL) (3 points). Sensitivity using WHO 4-symptom TB screen was 73%, 80%, 94%, and 94% in XPRES, XPHACTOR, TBFT, and Gugulethu cohorts, respectively, but increased to 88%, 87%, 97%, and 97%, when a clinical score of $\geq$2 was used. Negative predictive value (NPV) also increased 1%, 0.3%, 1.6%, and 1.7% in XPRES, XPHACTOR, TBFT, and Gugulethu cohorts, respectively, when the clinical score of $\geq$2 replaced WHO 4-symptom TB screen. Categorizing risk scores into low (<2), moderate (2 to 10), and high-risk categories (>10) yielded TB prevalence of 1%, 1%, 2%, and 6% in the lowest risk group and 33%, 22%, 26%, and 32% in the highest risk group for XPRES, XPHACTOR, TBFT, and Gugulethu cohorts, respectively. At clinical score $\geq$2, the number needed to screen (NNS) ranged from 5.0 in Gugulethu to 11.0 in XPHACTOR. Limitations include that the risk score has not been validated in resource-rich settings and needs further evaluation and validation in contemporary cohorts in Africa and other resource-constrained settings.

## Conclusions

The simple and feasible clinical score allowed for prioritization of sensitivity and NPV, which could facilitate reductions in mortality from undiagnosed TB and safer administration of TPT during proposed global scale-up efforts. Differentiation of risk by clinical score cutoff allows flexibility in designing differentiated HIV–TB care to maximize impact of available resources.

## Author summary

### Why was this study done?

- Tuberculosis (TB) remains the most common cause of death among people living with HIV (PLHIV) and is often undiagnosed at time of death.

- Rapid scale-up of tuberculosis preventive therapy (TPT) to 13 million PLHIV in low- and middle-income countries (LMICs) has been proposed for 2021; however, active TB is commonly asymptomatic and therefore missed by current WHO-recommended 4-symptom TB screening rules.

- Therefore, more sensitive TB screening tools are needed to better facilitate early TB diagnosis and safer scale-up of TPT to PLHIV by avoiding TPT prescription to clients with asymptomatic active TB, who need TB treatment.

## What did the researchers do and find?

- We derived a TB risk score for PLHIV from XPRES trial data and validated the score on 3 external datasets.

- We prioritized high sensitivity and ability to correctly rule out TB (i.e., high negative predictive value (NPV)) at key time points in care such as HIV clinic enrollment and before TPT prescription.

- Both logistic regression and random forest machine learning approaches were used to identify the 6 most important predictors, commonly available in LMIC clinic settings.

- In the external datasets, TB risk score ≥2 had higher sensitivity (87% to 97%) than WHO 4-symptom screening rule and increased NPV by 0.3% to 1.7%.

- Three risk groups were identified by the score, with active TB prevalence in external datasets ranging from 1% to 6% in the lowest to 22% to 32% in the highest risk groups.

## What do these findings mean?

- Following further validation, this clinical score could improve early detection of active TB to reduce morbidity and mortality from undiagnosed TB.

- Use of the clinical score cutoff of ≥2 during the proposed TPT scale-up for 13 million PLHIV could potentially avoid many thousands of PLHIV with active TB being inappropriately prescribed TPT.

- By differentiating 3 risk groups, the score also allows for the development of differentiated service delivery models suitable for LMIC.

## Introduction

Tuberculosis (TB) remains the most common cause of death among people living with HIV (PLHIV), with 251,000 HIV-associated TB deaths in 2018, over 95% of which occurred in low- and middle-income countries (LMICs) [1]. Among PLHIV who die from TB, TB often remains undiagnosed at the time of death [2,3]. The World Health Organization (WHO) recommends a 4-symptom TB screening rule (i.e., for cough, weight loss, night sweats, or fever) to determine which PLHIV need investigation for active TB and which are eligible for immediate tuberculosis preventive therapy (TPT) [4]. WHO 4-symptom TB screening rule is recommended for LMIC regardless of expected prevalence of active TB, setting (e.g., high or low TB incidence settings), or antiretroviral therapy (ART) status (ART-naive or ART-experienced) [4].

However, screening accuracy of WHO 4-symptom screening rule varies by population, setting, and ART status, raising the question whether a "one-size-fits-all" screening rule is appropriate. For example, a recent meta-analysis observed that while sensitivity of WHO 4-symptom TB screening rule is about 89% among ART-naive PLHIV, it is only 51% among people on ART due to a higher prevalence of subclinical TB (i.e., asymptomatic disease, caused by viable *Mycobacterium tuberculosis*, detectable by microbiologic tests or radiography [5]) among stable ART patients [6–8]. At a time when global health donors have committed to reaching over 13 million PLHIV on ART with TPT by 2021 [9], low sensitivity of WHO

4-symptom screening rule for active TB among PLHIV on ART warrants consideration of more sensitive screening approaches [10]. Although new WHO guidelines recommend adding chest radiography (CXR) to the screening rule for PLHIV on ART to increase sensitivity and negative predictive value (NPV), CXR is not available in many LMIC clinic settings; this comes at the expense of specificity, carries significant additional costs and operational challenges, and might hinder rather than expedite TPT scale-up in some LMIC settings [6,11]. Subclinical TB can also be present among severely immune compromised PLHIV [12] and among pre-ART patients without advanced disease in high prevalence settings [13], among whom failing to detect active TB can have serious health consequences for patients and impede disease control activities [14]. Finally, WHO 4-symptom screening rule does not allow TB risk differentiation into low-, moderate-, and high-risk groups, which might inform differentiated models of care.

Therefore, we aimed to develop a predictive clinical score based on variables commonly available in resource-constrained clinics, to define a range of cutoffs, with associated screening sensitivity, specificity, NPV, positive predictive value (PPV), percentage screened into diagnostic test algorithms, and number needed to screen (NNS) to detect one person with active TB.

## Methods

We used data from the Xpert Package Rollout Evaluation using a stepped wedge design (XPRES) trial conducted in Botswana to derive the predictive TB clinical score [15,16]. We split XPRES cohort data geographically into 11 southern and 11 northern clinics to serve as an internal derivation and validation dataset, respectively. We used 2 different but complementary modeling approaches to generate a parsimonious TB clinical risk score comprised of variables easily available in a resource-constrained clinic setting: (1) logistic regression models; and (2) random forest machine learning models. Random forest machine learning models are particularly useful for identifying important nonlinear associations between predictors and outcomes because the modeling approach does not rely on assumptions of average linear or curvilinear associations [17]. Having derived the clinical score, we then used data from 3 other settings to externally validate the derived clinical score.

Firstly, we used prospective cohort data for XPHACTOR study enrollees from Gauteng Province, South Africa (SA) [18]. XPHACTOR trial enrollees differed from XPRES enrollees in that 89% of enrollees were stable on ART at study enrollment versus 0% at XPRES study enrollment. Secondly, we used cluster randomized trial (CRT) data from the TB Fast Track (TBFT) trial from Gauteng, Limpopo, and North West Provinces in SA. TBFT trial enrollees differed from XPRES enrollees in that all had advanced HIV disease (all had CD4 count <150 cells/µL at study enrollment), but, similar to XPRES, no TBFT trial enrollees had started taking ART [19]. Thirdly, we used prospective cohort data from the Western Cape, SA (the Gugulethu cohort). Gugulethu cohort enrollees differed from XPRES enrollees in that Gugulethu has a very high background TB notification rate in the general population (>1,000/100,000 population [20]) compared with XPRES enrollees in Botswana where background TB notification rates were ≤500/100,000 [21], although similar to XPRES all Gugulethu cohort enrollees had not started ART at the time of study enrollment [22]. We compared screening accuracy of our derived clinical scores with existing WHO TB symptom screening criteria for active TB among PLHIV in each of these purposefully diverse populations.

### XPRES study design and participants for prediction tool development

XPRES was a stepped wedge CRT with a retrospective baseline component conducted at 22 health facilities, including 5 hospitals and 17 clinics, which were purposively selected to be

representative of HIV treatment clinics in Botswana [15]. In the prospective, stepped wedge portion of the trial, all nonincarcerated, consenting, HIV–positive persons not yet on ART, regardless of TB treatment or symptom status, presenting to the study clinics between August 2012 and end of March 2014, were eligible for enrollment. This analysis was limited to adolescents and adults (aged ≥12 years old), prospectively enrolled in the XPRES trial without a known TB diagnosis upon arrival at the study clinics (S1 Fig).

## XPRES procedures

Per Botswana national guidelines during the time period of the study, all XPRES study participants were eligible for ART initiation if they had a CD4 count ≤350 cells/μL, were diagnosed as having WHO stage III/IV, or were pregnant or breastfeeding [23]. All study participants received clinical care and follow-up appointments per Ministry of Health and Wellness guidelines, which included WHO TB symptom screening at the first and all subsequent clinic visits (S1 Table).

**Interventions.** The prospective XPRES cohort was recruited within 2 phases of the stepped wedge trial. In the first phase, all prospective XPRES participants received 2 enhanced care interventions in addition to standard of care (SOC): (1) additional support for intensified TB case finding; and (2) intensified tracing for patients missing clinic appointments. In the second phase, the Xpert MTB/RIF assay was initiated in place of sputum smear microscopy for TB diagnosis. We have previously shown that there was no significant difference in TB case finding between the first and second prospective phases of XPRES (5% versus 6%), although the prevalence of microbiologically confirmed TB was higher in the post-Xpert study phase (51% versus 65%) [16]. Details of the enrollment, intensified TB case finding and retention interventions, and follow-up procedures are described in a Supporting information appendix (S1 Text). XPRES participants were followed for 12 months, or until the end of TB treatment, whichever was later. The final follow-up visits for XPRES enrollees were in June 2015.

## Development and internal validation of the prediction model

A clinically useful prediction model should demonstrate accurate prediction of the outcome in data other than that in which the model was developed. Therefore, we split the XPRES dataset in an approximately 1:1 ratio into southern clinics (n = 11) and northern clinics (n = 11) with southern clinics serving as the derivation dataset and northern clinics as the model validation dataset. This nonrandom approach to splitting the dataset is preferable to randomly splitting patients into derivation and validation groups because it reduces the similarity of the 2 sets of patients and is a stricter test of the derived model [24].

**Outcome.** The outcome of interest for this analysis was new diagnosis of active TB (clinical or microbiologically confirmed), within 6 months of arrival at the HIV treatment clinic. In this manuscript, this outcome is referred to as undiagnosed prevalent TB at arrival at the HIV treatment clinic [18,25,26]. Active TB during this initial 6-month post-clinic enrollment period is considered prevalent rather than incident TB based on prior clinical cohort data showing that 87% of active TB cases identified in months 0 to 6 after HIV clinic enrollment could have been diagnosed at the HIV clinic enrollment visit [26]. In addition, data from Zimbabwe show that the mean duration of smear positivity prior to TB diagnosis among HIV–positive adults to be 18 to 33 weeks [25]. Precedent for this approach and definition have been previously published [18]. We implemented intensive efforts to ascertain true active TB disease among participants with TB case finding procedures previously published and provided in a Supporting information appendix (S1 Text) [15,16].

**Candidate predictor variables.** We selected candidate predictor variables for potential inclusion in the predictive models based on prior publications and the need for variables to be reproducible, objective, and readily available in resource-constrained clinic settings [27]. We considered variables known to be associated with active TB including age, sex, marital status, education level, employment status, previous/current work as a miner, smoking history, prior TB treatment, history of a TB contact in the last 24 months, presence and number of WHO TB symptoms, body mass index (BMI) (weight/height$^2$), hemoglobin concentration, CD4 count, temperature at ART initiation in degrees Celsius, and respiratory rate at enrollment visit [18,28–30].

**Logistic regression model approach.** Within the derivation dataset, we performed univariable logistic regression analyses assessing the association of each variable with risk of prevalent active TB. Continuous variables were assessed for nonlinearity with log odds of TB using multivariable fractional polynomial (MFP) models, as well as by comparing Akaike information criteria (AIC) and Bayesian information criteria (BIC) between models with linear or fractional polynomial terms. Where nonlinearity was observed, the appropriate fractional polynomial terms were included in the logistic regression. We also examined scatter plots of untransformed and transformed continuous variables and risk of TB to assess inflection points that might inform appropriate categorization of continuous variables. Where inflection points were close to previously published cutoffs for categorizing continuous variables, the previously published approach was used.

For the multivariable logistic regression analysis, a complete case analysis was chosen because few data (<10%) were missing. To inform generation of a parsimonious multivariable model, we used a stepwise backward elimination approach, starting with all candidate variables and excluding variables sequentially if $p > 0.01$ using both automatic and manual approaches. Prior regression derived scores used $p$-value cutoffs of >0.05 [18,31,32]; however, there is no accepted standard $p$-value cutoff for backward or forward stepwise variable elimination approaches [27]. Because we aimed to generate a parsimonious model, to increase feasibility of the practical clinical score in LMIC clinic settings, we used a >0.01 cutoff in line with recommendations from Royston and colleagues [27,33]. We also explored if findings changed using a forward stepwise addition approach. Where 2 or more predictors were highly correlated, only 1 was selected, to simplify the prognostic model. Plausible interactions between covariates (e.g., between sex and BMI [34]) were assessed using the likelihood ratio test.

**Random forest model approach.** We first built a random forest model with all 15 possible candidate variables that were included in the backward stepwise elimination approach. We fit the model using the randomForest R package with 1,000 trees. We used the *bestmtry* function to identify the optimum number of variables to be randomly included in each of the 1,000 trees (R version 3.6.1, R Core Team, 2017, R Foundation for Statistical Computing, Vienna, Austria) [35]. We used this model to order the 15 variables according to importance in predicting TB as measured by the mean decrease in Gini value for each variable [36]. The Gini value is a measure of decision tree node purity [37]. To develop a single decision tree, the best split at each node is assessed by evaluating which cutoff gives the most homogenous classifications (i.e., lowest Gini impurity according to published formulas [37]). The mean decrease in impurity is the average of a variable's total decrease in node impurity, weighted by the proportion of samples reaching that node in each individual tree in the random forest. Therefore, high mean decrease in Gini value indicates higher variable importance (i.e., the variable was on average important in splitting nodes into groups that had TB versus did not have TB).

We compared results with the logistic regression to assess if potentially important discriminatory variables had been eliminated in the backward stepwise regression. To assess any potentially important loss of discrimination through eliminating variables to create a parsimonious

model, and to assess potential differences in discriminatory capacity between model approaches, we compared area under the receiver operating characteristic (AUROC) curve values between modeling approaches (logistic regression versus random forest) and between variable selection approaches (models with all 15 variables selected versus parsimonious models). Information from the backward stepwise regression and random forest modeling was used to generate the final parsimonious model.

**Internal validation of parsimonious model.** In both the derivation and validation datasets, we assessed multivariable logistic regression model calibration graphically in a calibration plot [24] and statistically using the Hosmer–Lemeshow test. We also assessed discrimination using the AUROC values. AUROC values of 0.7 to 0.79, 0.8 to 0.89, and >0.9 are, respectively, considered acceptable, excellent, and outstanding discrimination [38].

**Clinical score generation.** The final multivariable model was used to generate a practical clinical score. For these models, continuous variables were categorized in a clinically meaningful manner based on their functional form and information from the published literature. Per published precedent, each beta coefficient from this logistic regression model was then rescaled to generate a clinical score by dividing each coefficient by the smallest positive model coefficient and rounding to the nearest integer [18,39]. The total number of points was summed for each participant to calculate their total clinical score. In a sensitivity analysis, we generated fully standardized coefficients using the *listcoef* command in STATA (Stata, 2009, Stata Statistical Software, Release 16, College Station, Texas, United States). We rescaled the standardized coefficients and assessed whether risk score discrimination varied between the scores derived from original beta coefficients versus fully standardized coefficients in XPRES, XPHACTOR, TBFT, and Gugulethu cohorts [40].

## External validation of risk scores

To externally validate the clinical risk score, we used data collected independently from the XPHACTOR cohort [18], TBFT trial [19], and Gugulethu cohort [22]. Characteristics of these studies as relate to setting, clinic types, eligibility criteria, dates of enrollment, ART eligibility criteria, TB symptoms screening, TB case finding approaches, and definition of active prevalent TB for this analysis are described in a Supporting information appendix (S2 Table).

For both the XPRES cohort (combined derivation and validation datasets) and the 3 validation datasets, we explored how sensitivity, specificity, PPV, NPV, percentage screened into diagnostic test algorithms, and AUROC curve values varied with increasing clinical score in terms of predicting active prevalent TB and compared this screening accuracy and discrimination performance with the current WHO TB symptom screening rule. Three risk groups were created to visualize increasing active prevalent TB risk with increasing clinical score and the percentage of ART enrollees falling into each risk group. The NNS to detect 1 TB case was compared between WHO TB symptom screening rules and a range of clinical score cutoffs.

All logistic regression and clinical score validation analyses were conducted using STATA 16 (Stata, 2009, Stata Statistical Software, Release 16). All random forest plot analyses and analyses to assess the mean decrease in Gini value associated with candidate predictor variables were done with R version 3.6.1. (R Core Team, 2017, R Foundation for Statistical Computing). Evaluating accuracy of different approaches to screening for undiagnosed active TB among PLHIV and evaluating prevalence and predictors of TB were part of the approved protocol (S1 Protocol). Augmenting the logistic regression approach with use of a machine learning approach and risk score generation were not prespecified but were used to build confidence in the analytic approach and translate the prespecified analysis into a screening tool potentially useful for clinicians. The study is reported in concordance with the Transparent Reporting of a

multivariable prediction model for Individual Prognosis or Diagnosis (TRIPOD) guidance for multivariable prediction models (S1 Checklist).

### Ethical review

Ethical approval for each of the source studies was obtained from the relevant ethics committees in the country of data collection and from the trial sponsors. All participants provided informed written consent, or where the enrollee could not read or write, witnessed verbal informed consent. Ethical approvals for XPRES were obtained from the US Centers for Disease Control and Prevention (CDC) Institutional Review Board (IRB), the Health Research and Development Division of the Health Research and Development Committee (HRDC) in Botswana, and the University of Pennsylvania IRB No.4. XPHACTOR was approved by the ethics committees at the University of the Witwatersrand, University of Cape Town, and the London School of Hygiene & Tropical Medicine. TBFT was approved by the research ethics committees of the University of the Witwatersrand and the London School of Hygiene & Tropical Medicine and the South African Medicines Control Council. The Gugulethu prospective cohort study was approved by the research ethics committees of the University of Cape Town and the London School of Hygiene & Tropical Medicine.

## Results

From the XPRES cohort, 5,418 eligible adult ($\geq$12 years old) study enrollees with complete data for candidate predictors were included in the analysis (Fig 1). Overall, 318 (6%) of 5,418 enrollees had undiagnosed prevalent active TB at HIV clinic registration and study enrollment. From this XPRES cohort, the internal derivation ($N$ = 2,771) and validation ($N$ = 2,647) datasets were created (Table 1). Key characteristics including median age (34 years), percentage female (67% to 68%), median CD4 (240 to 249 cells/μL), and prevalence of active TB at enrollment (5% to 7%) were similar between XPRES derivation and validation datasets (Table 1).

### Variable importance in logistic regression and random forest models

Table 2 summarizes the results of univariable and multivariable logistic regression model development. Although age (linear continuous variable), education level achieved, prior/current work as a miner, previous TB treatment, and respiratory rate (transformed variable) were associated with prevalent TB in univariable analysis, these variables were eliminated in the stepwise backward elimination approach due to $p$-values in multivariable analysis >0.01. The full 15-variable multivariable logistic regression model is in a Supporting information appendix (S3 Table). Both backward stepwise elimination and forward stepwise addition approaches yielded the same results in terms of final variables selected, adjusted odds ratios, and $p$-values.

Rankings of variable importance as measured by size of beta coefficient in the logistic regression model (S4 Table) and by mean decrease in Gini value in the random forest model (S2 Fig) were similar with respect to presence or absence of WHO TB symptoms, temperature, and BMI, with these variables among the most important predictors. Notably, the transformed term of BMI, which was eliminated in the backwards stepwise logistic regression at $p$ = 0.408, was considered the second most important variable according to the mean decrease in Gini value approach (52.766) and third most important in terms of size of the logistic regression beta coefficient (1.617). Given the importance of BMI in the variable ranking approach, importance in the published literature, and availability in resource-constrained clinics, BMI was retained in final multivariable model (Table 2). The final multivariable model included sex,

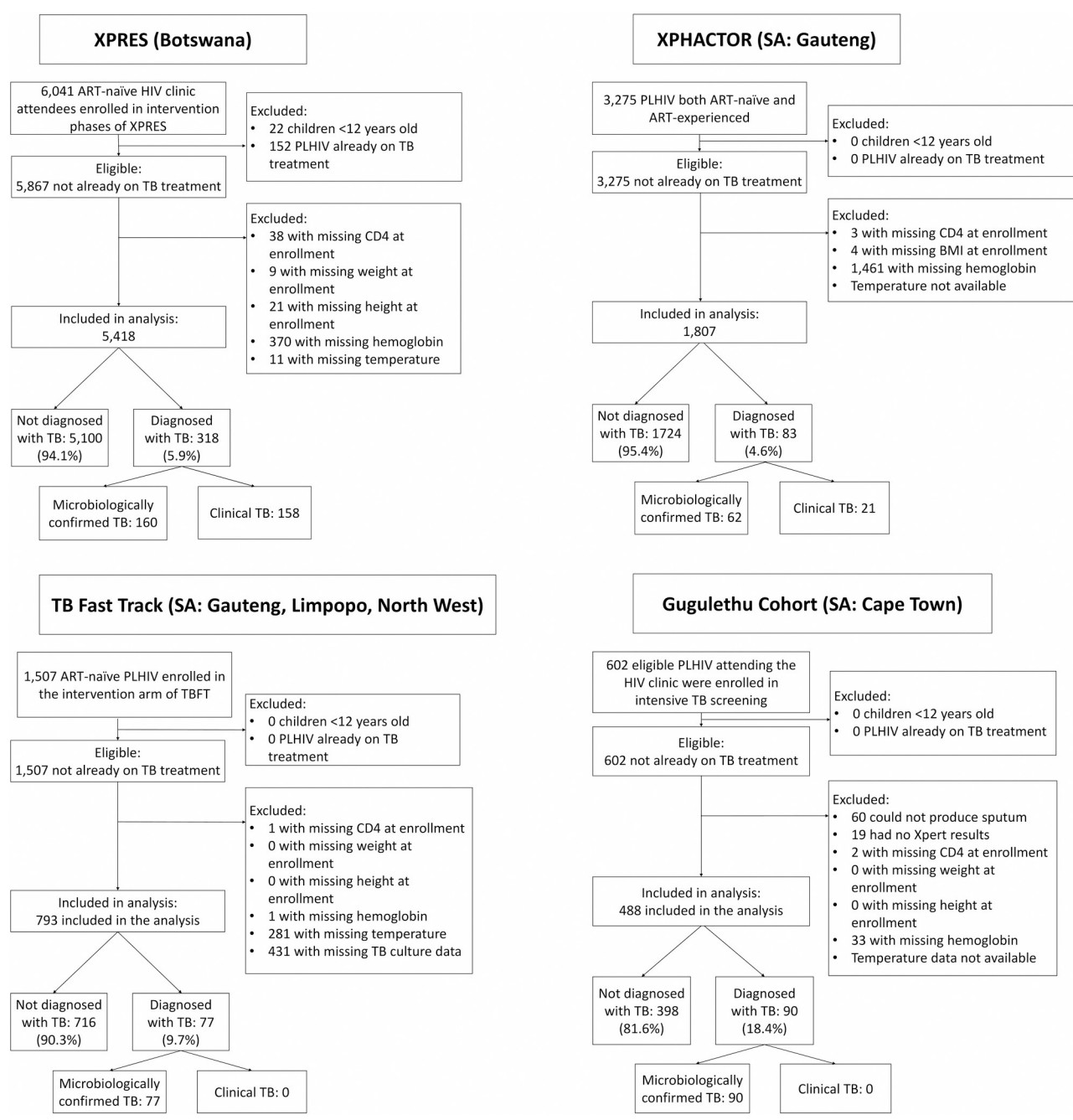

**Fig 1. Study profiles for the derivation and validation datasets.** ART, antiretroviral therapy; PLHIV, people living with HIV; SA, South Africa; TB, tuberculosis; TBFT, TB Fast Track.

smoking history, presence/absence of ≥1 WHO TB symptom, BMI (as a transformed term per the MFP analysis), temperature (modeled as 2 transformed terms per the MFP analysis), and hemoglobin concentration (continuous, linear term) (Table 2). There was no substantial correlation between measured temperature in its transformed or linear form and reported fever (Pearson correlation coefficient (r) = 0.1316) or between measured temperature and ≥1 reported WHO TB symptom (r = 0.0846).

**Table 1. Comparison of derivation and validation datasets (internal and external).***

| | | | Derivation dataset (Botswana southern clinics: N = 2,771) | | Validation dataset (Botswana northern clinics: N = 2,647) | | External validation dataset (SA, XPHACTOR: N = 1,807) | | External validation dataset (TBFT, SA: N = 793) | | External validation dataset (Gugulethu cohort, CT, SA: N = 488) | |
|---|---|---|---|---|---|---|---|---|---|---|---|---|
| **Demographics** | | | **n** | **(Median or %)** | **n** | **(Median or %)** | **n** | **(Median or %)** | **n** | **(Median or %)** | **n** | **(Median or %)** |
| | Age (years) n, median (IQR) | | 2,771 | 34.3 (28.8 to 41.3) | 2,647 | 33.5 (28.3 to 40.8) | 1,807 | 40.0 (34.0 to 47.0) | 793 | 38.0 (32.0 to 45.0) | 488 | 33.6 (27.9 to 40.7) |
| | Female, n, % | | 1,862 | 67% | 1,790 | 68% | 1,290 | 71% | 424 | 53% | 310 | 64% |
| | If female, pregnant, n, % | | 499 | 27% | 568 | 32% | 0 | 0% | 0 | 0% | 0 | 0% |
| | Marital status, n, % | Married/civil union | 306 | 11% | 242 | 9% | | | | | | |
| | | Single | 2,353 | 85% | 2,322 | 88% | | | | | | |
| | | Widowed/divorced | 112 | 4% | 83 | 3% | | | | | | |
| | Smoking history (ever), n, % | | 466 | 17% | 575 | 22% | 388 | 21% | 180 | 23% | 185 | 38% |
| | Employed, n, % | | 1,467 | 53% | 1,023 | 39% | | | | | | |
| | Education, n, % | None | 154 | 6% | 235 | 9% | | | | | | |
| | | Primary | 637 | 23% | 614 | 23% | | | | | | |
| | | Secondary | 1,689 | 61% | 1,597 | 60% | | | | | | |
| | | Higher | 291 | 11% | 201 | 8% | | | | | | |
| | Ever a miner, n, % | | 124 | 4% | 143 | 5% | | | | | | |
| **HIV/TB history** | | | | | | | | | | | | |
| | Taking ART at study enrollment | | 0 | 0% | 0 | 0% | 1,612 | 89% | 0 | 0% | 0 | 0% |
| | Previous TB treatment, n, % | | 232 | 8% | 169 | 6% | | | | | 130 | 27% |
| | TB contact in last 24 months, n, % | | 266 | 10% | 230 | 9% | | | | | | |
| | WHO TB symptoms, n, % | | | | | | | | | | | |
| | | Cough | 533 | 19% | 466 | 18% | 364 | 20% | 424 | 53% | 243 | 50% |
| | | Weight loss | 533 | 19% | 577 | 22% | 243 | 13% | 621 | 78% | 331 | 68% |
| | | Fever | 262 | 9% | 223 | 8% | 105 | 6% | 269 | 34% | 139 | 28% |
| | | Night sweats | 257 | 9% | 243 | 9% | 135 | 7% | 297 | 37% | 199 | 41% |
| | Number of WHO TB symptoms, n, % | 0 | 1,979 | 71% | 1,837 | 69% | 1,299 | 72% | 133 | 17% | 67 | 14% |
| | | 1 | 349 | 13% | 397 | 15% | 384 | 21% | 173 | 22% | 127 | 26% |
| | | 2 | 200 | 7% | 203 | 8% | 136 | 8% | 191 | 24% | 135 | 28% |
| | | 3 | 136 | 5% | 134 | 5% | 41 | 2% | 145 | 18% | 121 | 25% |
| | | 4 | 107 | 4% | 76 | 3% | 17 | 1% | 151 | 19% | 38 | 8% |
| | Duration of WHO symptoms | n, median (IQR) | 792 | 60 (30 to 120) | 810 | 60 (21 to 150) | | | | | | |
| **Clinical characteristics** | | | | | | | | | | | | |
| | CD4+ T-cell count (cells/µL) | n, median (IQR) | 2,771 | 240 (131 to 314) | 2,647 | 249 (151 to 321) | 1,807 | 400 (246 to 600) | 793 | 73 (34 to 109) | 488 | 167 (95 to 231) |
| | Weight (kg)** | n, median (IQR) | 2,771 | 59 (52 to 69) | 2,647 | 60 (53 to 69) | | | 793 | 57 (50 to 66) | 488 | 64 (56 to 73) |
| | BMI (kg/m²) | n, median (IQR) | 2,771 | 21.8 (19.2 to 25.4) | 2,647 | 21.5 (18.9 to 24.7) | 1,807 | 25.0 (21.4 to 29.3) | 793 | 20.9 (18.6 to 24.5) | 488 | 23.5 (20.9 to 27.1) |
| | Hemoglobin g/dL | n, median (IQR) | 2,771 | 11.9 (10.5 to 13.3) | 2,647 | 12.0 (10.7 to 13.4) | 1,807 | 13.1 (11.8 to 14.3) | 793 | 11.1 (9.6 to 12.8) | 488 | 12.0 (10.6 to 13.4) |
| | Temperature (˚C) | n, median (IQR) | 2,771 | 36.2 (35.8 to 36.7) | 2,647 | 36.1 (35.7 to 36.5) | | | 793 | 36.3 (36.0 to 36.6) | | |
| | Respiratory rate (breaths/min) | n, median (IQR) | 2,771 | 20 (18 to 21) | 2,647 | 18 (17 to 20) | | | | | | |
| **New TB diagnosis** | | | | | | | | | | | | |
| | Cumulative prevalent active TB, n, % | | 189 | 6.8% | 129 | 4.9% | 83 | 4.6% | 77 | 9.7% | 90 | 18.4% |
| | Cumulative incidence microbiologically confirmed TB, n, % | | 96 | 3.5% | 64 | 2.4% | 62 | 3.4% | 77 | 9.7% | 90 | 18.4% |
| | Time to diagnosis of prevalent TB (days) | n, median (IQR) | 189 | 16 (7 to 35) | 129 | 19 (4 to 48) | | | | | | |

* Where variable is blank, the data were not collected or not provided from the source study for this analysis.

** BMI was used as the covariate for nutritional status rather than weight (weight was not considered as an independent predictor).

BMI, body mass index; CT, Cape Town; IQR, interquartile range; SA, South Africa; TB, tuberculosis; TBFT, TB Fast Track; WHO, World Health Organization.

**Table 2. Univariable and multivariable logistic regression analysis in the derivation dataset (N = 2,771).**

| | | Not diagnosed with TB (N = 2,582) | | Diagnosed with TB within 6 months (N = 189) | | Unadjusted | | | Final adjusted regression | | |
|---|---|---|---|---|---|---|---|---|---|---|---|
| | | n | Median(IQR)/% | n | Median(IQR)/% | OR | 95% CI | p-value | AOR | 95% CI | p-value |
| **Demographics** | | | | | | | | | | | |
| Age, years (for every 10-year increase) | | | 34 (29 to 41) | | 38 (32 to 44) | 1.24 | (1.15 to 1.33) | <0.001 | | | |
| Sex | Female | 1,768 | 95% | 94 | 5% | 1.00 | -- | -- | 1.00 | -- | -- |
| | Male | 814 | 90% | 95 | 10% | 2.20 | (1.63 to 2.95) | <0.001 | 1.91 | (1.27 to 2.88) | 0.002 |
| Marital status | Married/civil union | 288 | 94% | 18 | 6% | 1.00 | -- | -- | | | |
| | Single | 2,190 | 93% | 163 | 7% | 1.19 | (0.71 to 2) | 0.508 | | | |
| | Widowed/divorced | 104 | 93% | 8 | 7% | 1.23 | (0.44 to 3.41) | 0.690 | | | |
| Smoking history (ever smoked) | No | 2,165 | 93% | 154 | 7% | 1.00 | -- | -- | 1.00 | -- | -- |
| | Yes—ever smoked | 417 | 89% | 50 | 11% | 1.82 | (1.56 to 2.12) | <0.001 | 1.44 | (1.12 to 1.85) | 0.004 |
| Employed | Employed | 1,370 | 93% | 97 | 7% | 1.00 | -- | -- | | | |
| | Unemployed | 1,212 | 93% | 92 | 7% | 1.07 | (0.71 to 1.62) | 0.742 | | | |
| Education | None | 137 | 89% | 17 | 11% | 1.00 | -- | -- | | | |
| | Primary | 588 | 92% | 49 | 8% | 0.67 | (0.38 to 1.18) | 0.166 | | | |
| | Secondary | 1,576 | 93% | 113 | 7% | 0.58 | (0.35 to 0.95) | 0.030 | | | |
| | Higher | 281 | 97% | 10 | 3% | 0.29 | (0.14 to 0.58) | <0.001 | | | |
| Ever a miner | No | 2,478 | 94% | 169 | 6% | 1.00 | -- | -- | | | |
| | Yes | 104 | 84% | 20 | 16% | 2.82 | (2.04 to 3.9) | <0.001 | | | |
| **HIV/TB history** | | | | | | | | | | | |
| Previous TB treatment | No | 2,381 | 94% | 158 | 6% | 1.00 | -- | -- | | | |
| | Yes | 201 | 87% | 31 | 13% | 2.32 | (1.43 to 3.78) | 0.001 | | | |
| Any TB contact in last 24 months | No | 2,339 | 93% | 180 | 7% | 1.00 | -- | -- | | | |
| | Yes | 243 | 91% | 24 | 9% | 1.33 | (0.83 to 2.14) | 0.233 | | | |
| Number of WHO symptoms | 0 | 1,936 | 98% | 43 | 2% | 1.00 | -- | -- | | | |
| | > = 1 | 646 | 82% | 146 | 18% | 10.18 | (6.79 to 15.26) | <0.001 | 6.91 | (4.55 to 10.49) | <0.001 |
| **Clinical characteristics** | | | | | | | | | | | |
| CD4 (per 10-cell increase) | | 2,582 | 247 (139 to 316) | 189 | 151 (57 to 255) | 0.96 | (0.94 to 0.97) | <0.001 | | | |
| Weight (per 1-kg increase)[a] | | 2,582 | 59.4 (52.3 to 69.2) | 189 | 53.7 (47.0 to 62.0) | 0.97 | (0.95 to 0.99) | 0.001 | | | |
| BMI (per 1-unit increase)[b] | | 2,582 | 21.9 (19.4 to 25.5) | 189 | 19.4 (17.2 to 22.3) | 0.90 | (0.83 to 0.97) | 0.004 | 0.98 | (0.93 to 1.05) | 0.612 |
| Hemoglobin (per 1g/dL increase) | | 2,582 | 12.0 (10.6 to 13.3) | 189 | 10.6 (9.2 to 12.3) | 0.76 | (0.69 to 0.83) | <0.001 | 0.78 | (0.7 to 0.86) | <0.001 |
| Temperature at enrollment (per 1˚C increase)[c] | | 2,582 | 36.2 (35.8 to 36.6) | 189 | 36.4 (36.0 to 37.1) | 2.13 | (1.57 to 2.88) | <0.001 | 1.46 | (1.18 to 1.81) | <0.001 |
| RR (breaths/min)[d] | | 2,582 | 20 (18 to 20) | 189 | 20 (18 to 22) | 1.03 | (1.01 to 1.05) | 0.010 | | | |

[a] Due to correlation between weight and BMI (r = 0.8837), weight was not included in the stepwise backward regression, because BMI is a better measure of nutritional status than weight alone.

[b] Due to nonlinearity in the association between BMI and log odds TB, BMI was modeled as a transformed term from the MFP analysis (transformed BMI = $X^{-.5}$ −.666749355, where X = BMI/10). Output shown is for the single linear term to facilitate interpretation of average BMI effect (i.e., higher BMI associated with lower TB risk). In the backward stepwise regression, the p-value associated with BMI term was 0.4077 at point of elimination. Given the importance of BMI as a predictor in the random forest model (second most important predictor), ease of availability of this variable in almost all resource-limited clinics, and importance of BMI in published literature, BMI was retained in the final adjusted model.

[c] Due to nonlinearity in the association between temperature and log odds TB, temperature was modeled as 2 transformed terms (term 1 = temperature -36.12674419; term 2 = $temperat^2$ −1305.141645). Output shown is for the single linear term to facilitate interpretation of average temperature effect (i.e., higher temperature associated with higher TB risk). In the backward stepwise regression, the p-value associated with each transformed term was 0.005 and 0.004, respectively.

[d] Due to nonlinearity in the association between RR and log odds TB, RR was modeled as a transformed term from the MFP analysis (transformed term = $X^{-1}$ −5.170636738, where X = RR/100). Output shown is for the single linear term to facilitate interpretation of average RR effect (i.e., higher RR associated with higher TB risk). In the backward stepwise regression, the p-value associated with the transformed term was 0.0251 at the point of elimination from the model.

AOR, adjusted odds ratio; BMI, body mass index; CI, confidence interval; IQR, interquartile range; MFP, multivariable fractional polynomial; OR, odds ratio; RR, respiratory rate; TB, tuberculosis; WHO, World Health Organization.

### Internal validation of final multivariable regression model

For the derivation dataset, the Hosmer–Lemeshow statistic for the TB prediction model ($p = 0.135$) (S5 Table), and the calibration curve (Fig 2), indicated good model fit. Although the Hosmer–Lemeshow statistic for the internal validation dataset ($p = 0.0001$) indicated lack of fit with overestimation of prevalent TB, with 169 cases of prevalent TB predicted versus 129 observed, (1) the Hosmer–Lemeshow test is sensitive to sample size, and our sample size is large; and (2) the calibration curve (Fig 2) indicated adequate prediction performance for the 10 risk groups. In addition, the AUROC curve values for the derivation (0.839; 95% CI, 0.811 to 0.868) and validation datasets (0.799; 95% CI, 0.757 to 0.841) indicated excellent and borderline excellent discrimination, respectively (Fig 2).

### Comparison of regression and random forest discrimination

Comparison of discriminatory performance between 15-covariate and 6-covariate parsimonious models (S3 Fig) indicated very little loss of discrimination by eliminating 9 of the covariates from the predictive model, building confidence in the final multivariate model. Similarly, although the random forest approach had far superior discrimination on the derivation dataset

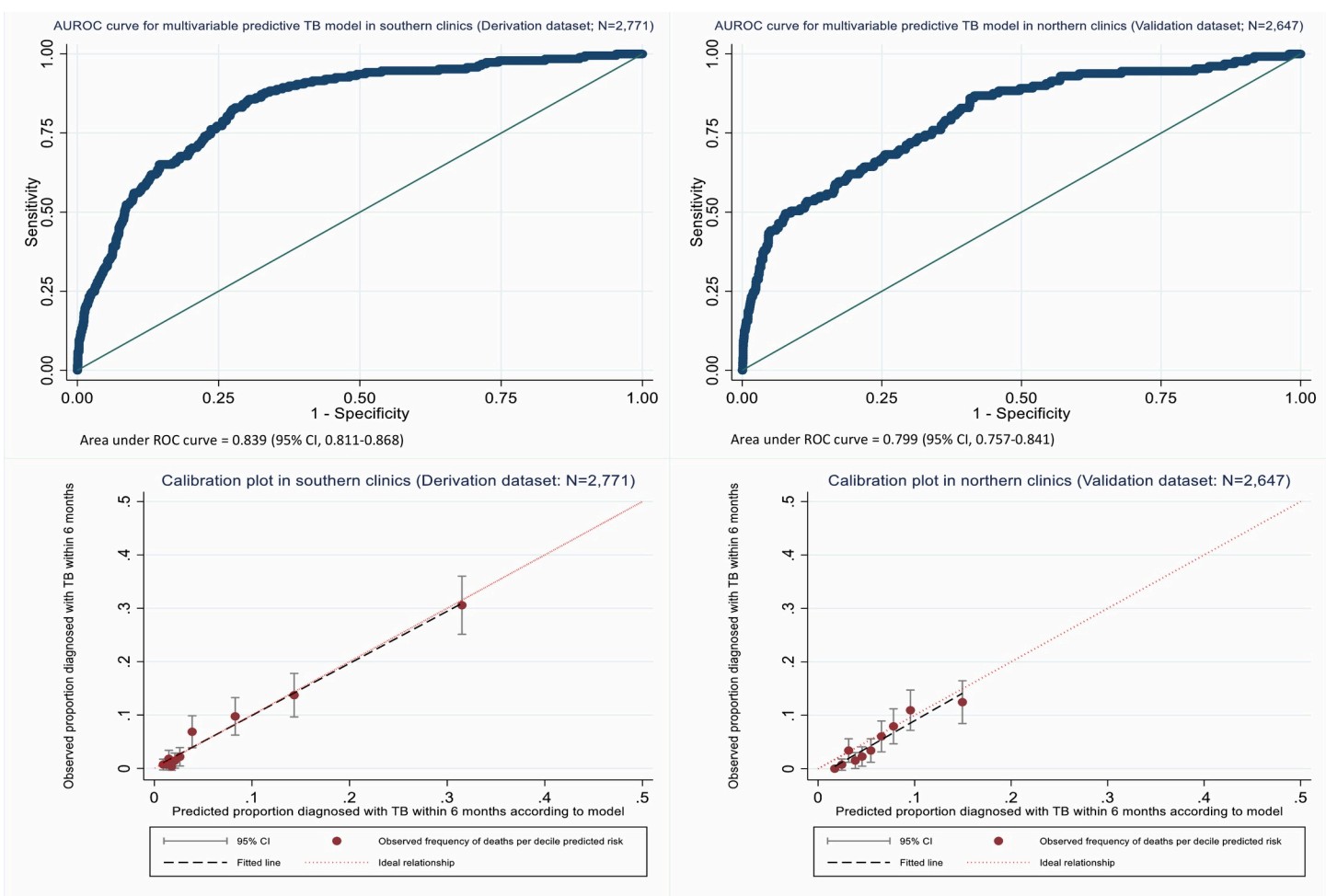

**Fig 2. Logistic regression model AUROC curves and calibration curves for the internal derivation and validation datasets, respectively.** AUROC, area under the receiver operating characteristic; TB, tuberculosis.

versus the logistic regression approach, both modeling approaches had similar discrimination in the internal validation dataset (S3 Fig).

## Transformation from regression model to clinical score

We used WHO-recommended cutoffs for severe anemia in adults (<8.0 g/dL) [41] and for being underweight (BMI<18.5 kg/m$^2$) to categorize hemoglobin and BMI variables, respectively. Temperature was classified as ≤37.5˚C versus >37.5˚C based on the observed distribution of TB prevalence risk as measured temperature increased and a common definition of a low-grade fever or higher (>37.5˚C) (S4 Fig) [42]. The multivariable model with categorization of these continuous variables in the derivation dataset is presented in Table 3.

The final model, categorized in this way, retained excellent discrimination in the derivation dataset (AUROC 0.823; 95% CI, 0.793 to 0.853) and acceptable discrimination in the validation dataset (AUROC 0.771; 95% CI, 0.727 to 0.815), and the Hosmer–Lemeshow statistic *p*-values were 0.1940 in the derivation and 0.0002 in the validation datasets indicating similar goodness of fit as was observed prior to variable categorization. The clinical scores that could be used in clinic settings to identify those at risk of prevalent active TB are illustrated in Fig 3.

## External validation of risk scores

The clinical score for each predictor was generated and applied to each external dataset, where the possible range for the total score was 0 to 20 (S6 Table). Performance of the clinical score at different cutoffs, in terms of sensitivity, specificity, NPV, PPV, and the percentage of clinic enrollees that would be offered a TB test is provided in a Supporting information appendix (S5 Fig). Across the 4 datasets, a clinical score of ≥7 would give similar sensitivity and specificity to WHO 4-symptom TB screening rule. Moving the clinical score to ≥2 would give superior sensitivity versus WHO 4-symptom TB screening rule, but with some loss of specificity. For example, sensitivity in detecting prevalent active TB using WHO 4-symptom TB screening rule was 73%, 80%, 94%, and 94% in XPRES, XPHACTOR, TBFT, and Gugulethu cohorts, respectively, but this increased to 88%, 87%, 97%, and 97%, when a clinical score of ≥2 was used. However, specificity would decline from 73%, 70%, 18%, and 16% if WHO 4-symptom TB-screen was used to 55%, 58%, 13%, and 12% if the clinical score of ≥2 was used. Similarly,

**Table 3. Multivariable model and clinical score in the derivation dataset (*N* = 2,771).**

| Predictor | | AOR | 95% CI | *p*-value | β coefficient | Score |
|---|---|---|---|---|---|---|
| WHO TB symptoms | No symptoms | 1.00 | -- | -- | | |
| | ≥1 symptom | 7.00 | (4.66 to 10.52) | <0.001 | 1.95 | 7 |
| Sex | Female | 1.00 | -- | -- | | |
| | Male | 1.35 | (0.88 to 2.08) | 0.173 | 0.30 | 1 |
| Smoker | Never | 1.00 | -- | -- | | |
| | Ever smoked | 1.32 | (1.03 to 1.7) | 0.030 | 0.28 | 1 |
| Hemoglobin | ≥8 g/dL | 1.00 | -- | -- | | |
| | <8 g/dL | 2.50 | (1.28 to 4.85) | 0.007 | 0.91 | 3 |
| Temperature | ≤37.5 | 1.00 | -- | -- | | |
| | >37.5 | 5.53 | (3.5 to 8.72) | <0.001 | 1.71 | 6 |
| BMI | ≥18.5 | 1.00 | -- | -- | | |
| | <18.5 | 1.70 | (1.12 to 2.59) | 0.013 | 0.53 | 2 |

AOR, adjusted odds ratio; BMI, body mass index; CI, confidence interval; TB, tuberculosis; WHO, World Health Organization.

| Risk Factor | Category | Associated points | Assigned score |
|---|---|---|---|
| Sex | Female | 0 | |
| | Male | 1 | ✚ |
| No. of WHO TB symptoms (cough, fever, weight loss, night sweats) | Zero | 0 | |
| | ≥1 | 7 | ✚ |
| Smoker | Never | 0 | |
| | Ever smoked | 1 | ✚ |
| Temperature (˚C) | ≤37.5 | 0 | |
| | >37.5 | 6 | ✚ |
| BMI (kg/m$^2$) | ≥18.5 | 0 | |
| | <18.5 | 2 | ✚ |
| Hemoglobin Level (g/dL) | ≥8.0 | 0 | |
| | <8.0 | 3 | ═ |
| Total | | | |

| <2 = low risk | 2–10 = moderate risk | >10 = high risk |
|---|---|---|

**Fig 3. Clinical score for predicting TB among PLHIV.** BMI, body mass index; PLHIV, people living with HIV; TB, tuberculosis; WHO, World Health Organization.

the percentage of patients screened into a TB diagnostic test algorithm (referred to as "screen in" in S5 Fig) per WHO 4-symptom TB screening rule would be 30%, 32%, 83%, and 86% in the XPRES, XPHACTOR, TBFT, and Gugulethu cohorts, respectively, but this increases to 45%, 42%, 87%, and 88% if a clinical score of ≥2 is used.

Notably, when the XPHACTOR dataset was restricted to clients on ART for >3 months, the clinical score retained good sensitivity and moderate specificity (S6 Fig). For example, at clinical score ≥2, sensitivity was 80% and specificity 60% versus WHO 4-symptom screening criteria that provided 69% sensitivity and 72% specificity.

The NPV of WHO 4-symptom TB screen was 97.7% and 98.6%, 96.2%, and 92.5% in the XPRES, XPHACTOR, TBFT, and Gugulethu cohorts, increasing to 98.7%, 98.9%, 97.8%, and 94.2% when the clinical score at cutoff ≥2 was used, reflecting a 1%, 0.3%, 1.6%, and 1.7% increase in NPV.

When restricting the XPRES and TBFT cohorts to those who died within 6 months of clinic enrollment, the clinical score at a cutoff of ≥2 had superior sensitivity to WHO 4-symptom TB screen in predicting TB in the XPRES cohort (94% versus 79%) and similar sensitivity in the TBFT cohort (100% versus 100%) (S7 Fig). However, specificity of the clinical score at ≥2 was inferior to that of WHO 4-symptom TB screen in both XPRES (16% versus 31%) and TBFT (3% versus 8%) cohorts.

Overall, the clinical score had superior discrimination in the XPRES and XPHACTOR datasets than in the TBFT and Gugulethu cohorts (S8 Fig). The XPRES and XPHACTOR cohorts were more similar with respect to median baseline CD4 count (245/μL in XPRES and 400/μL in XPHACTOR) compared with TBFT and Gugulethu cohorts (73/μL in TBFT and 167/μL in Gugulethu cohorts) (Table 1). Similarly, XPRES and XPHACTOR cohorts were more similar with respect to baseline prevalence of active prevalent TB (6% in XPRES and 5% in XPHACTOR) compared with TBFT and Gugulethu cohorts (10% in TBFT and 18% in Gugulethu cohort).

In 4 sensitivity analyses, we compared discrimination of our final risk score, derived from the multivariable model beta coefficients described in Table 3 versus (1) risk scores derived from fully standardized beta coefficients; (2) risk scores excluding BMI (the variable identified as important by random forest model variable ranking); (3) risk scores excluding hemoglobin (the only variable requiring a blood test); and (4) risk scores excluding sex and smoking (variables assigned only 1 point in the risk score). In all sensitivity analyses, our final risk score had modest but consistently superior discrimination across XPRES, XPHACTOR, TBFT, and Gugulethu cohorts (S7 Table).

Risk scores were grouped into low (<2), moderate (2 to 10), and high-risk categories (>10) (Fig 4). Prevalence of active TB among enrollees in low-, moderate-, and high risk groups was 1%, 3%, and 33% among XPRES enrollees, 1%, 11%, and 22% among XPHACTOR enrollees, 2%, 8%, and 26% for TBFT enrollees, and 6%, 19%, and 32% for Gugulethu cohorts, respectively, indicating a differentiation of prevalent TB risk by the respective clinical scores.

## NNS to diagnose one TB case

In the cohorts with the highest prevalence of active TB (TBFT and Gugulethu), a clinical score with cutoff of ≥2 would give a marginally higher NNS to diagnose one TB case compared with the 4-symptom WHO screen (Fig 5); the NNS increased from 8.6 to 9.3 in TBFT and from 4.7 to 5.0 in Gugulethu cohorts. In contrast, in cohorts with lower prevalence of active prevalent TB (XPPRES and XPHACTOR), the NNS increased to a larger extent (from 5.0 to 9.3 in XPRES and from 6.9 to 11.0 in XPHACTOR). The NNS in the highest risk group with clinical score >10 (i.e., equivalent to ≥11 in Fig 5) was uniformly low being 3.0, 4.5, 3.9, and 3.1 in XPRES, XPHACTOR, TBFT, and Gugulethu cohorts, respectively. If the NNS threshold was set at about 5.0, this would correspond to clinical scores of about ≥9 to 10 in XPRES, XPHACTOR, and TBFT cohorts, but ≥2 in the Gugulethu cohort.

## Discussion

To our knowledge, this study provides new information by deriving and externally validating an initial clinical score for active TB among both ART-naive and ART-experienced adult PLHIV that includes but does not rely solely on WHO TB symptom screening and allows flexibility in choosing the desired sensitivity, specificity, NPV, PPV, and NNS across a range of cutoffs, depending on the setting, use case scenario, and population served. In addition, following further validation and evaluation steps, the screening tool could potentially be used to reduce the likelihood of missing subclinical TB, which accounted for 6% to 27% of all TB cases across

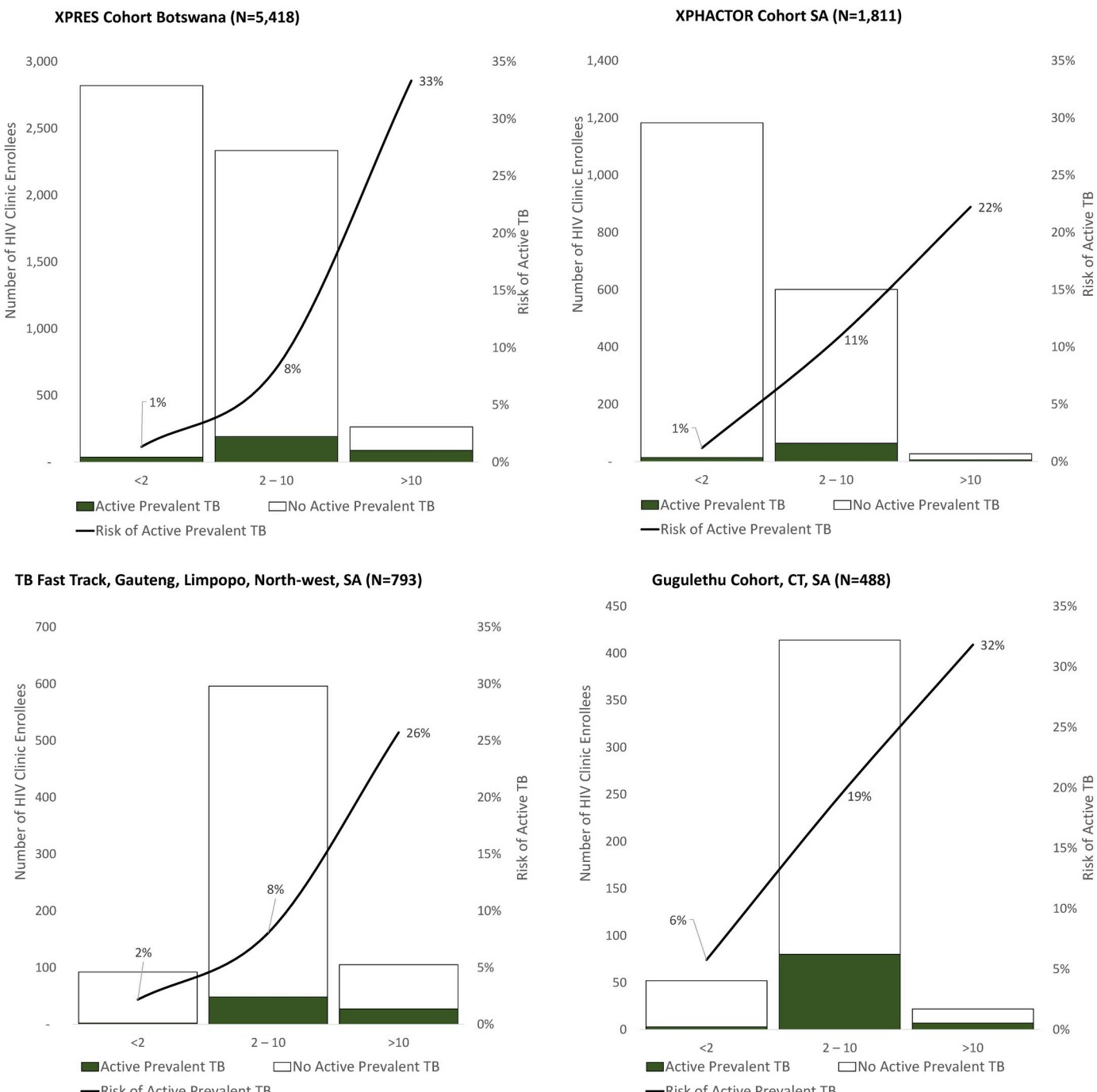

**Fig 4. TB risk stratification into low-, moderate-, and high-risk groups by study cohort.** CT, Cape Town; SA, South Africa; TB, tuberculosis.

studies; this could potentially help reduce morbidity and mortality due to late or missed TB diagnosis and reduce TPT prescription to PLHIV needing a full TB treatment course. Similarly, following further validation efforts, the screening tool's differentiation of 3 risk groups could be used to inform differentiated care in LMIC clinic settings, which could potentially improve efficiency and impact morbidity and mortality. Finally, the different modeling

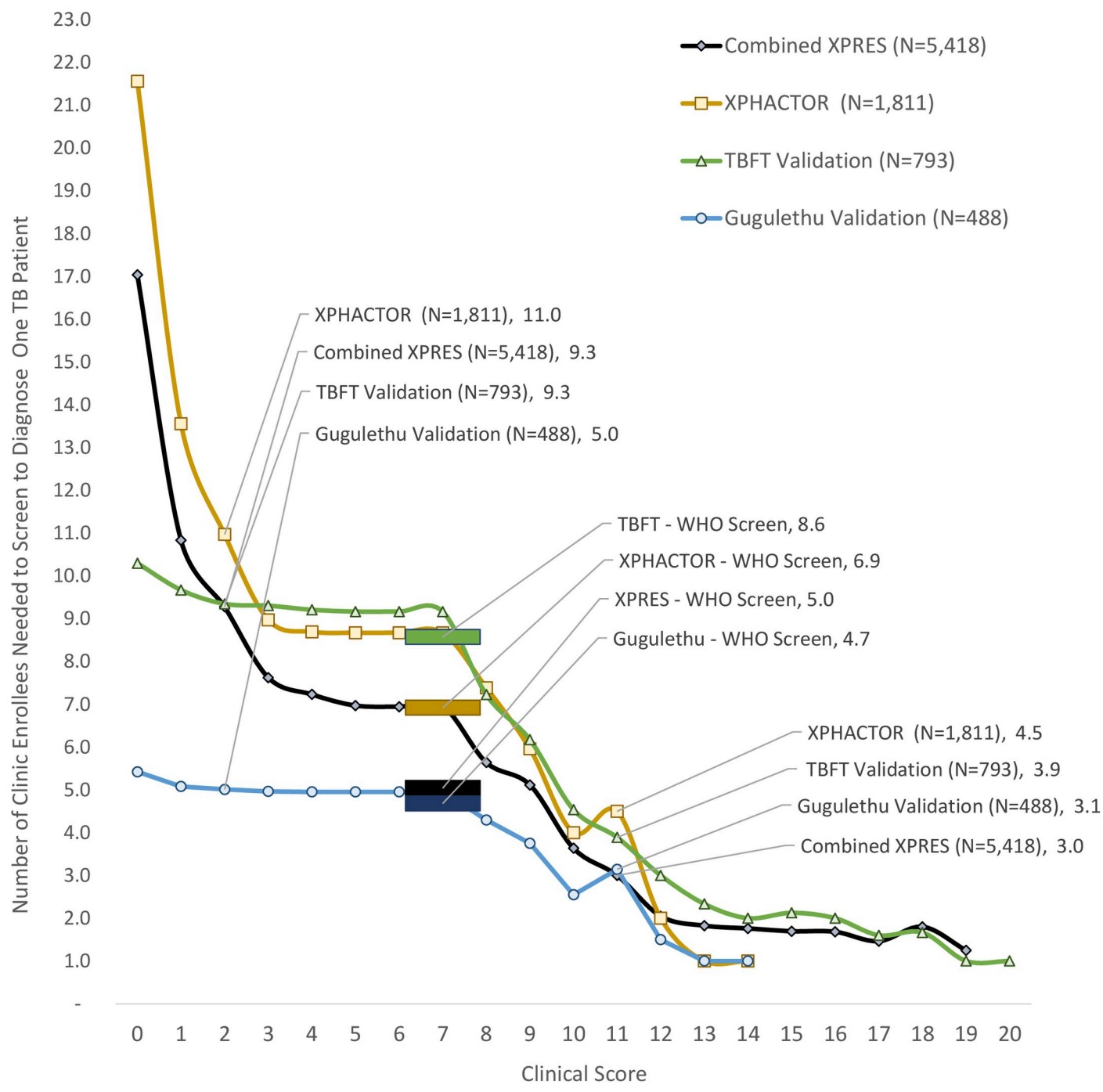

**Fig 5. NNS to detect one case of active TB by clinical score cutoff and by study cohort.** NNS, number needed to screen; TB, tuberculosis; TBFT, TB Fast Track; WHO, World Health Organization.

approaches provide unique insight into covariate predictor importance and practical ways machine learning can be helpful in predicting TB.

While 5 previous studies have generated clinical scores for TB among PLHIV, 3 were designed as a second step after screening positive using WHO 4-symptom TB screening rule [18,32,43], and 2 generated a relatively complex score (13 signs and symptoms), were focused

on ART-naive patients only in Bissau, and lack external validation [44,45]. One recent TB clinical score was derived from a cohort of HIV–positive and HIV–negative patients in SA who had symptoms of active TB; however, this score would not be able to detect asymptomatic active TB among PLHIV [46]. Our tool was deliberately designed to use widely available variables in LMIC settings and was externally validated using 3 different cohorts. The only blood test needed for our score is hemoglobin concentration. Point-of-care (POC) hemoglobin measurement devices are widely available, durable, easy to use, have good accuracy [47,48], are useful for non-HIV–related care, and are inexpensive [49]. Noninvasive transcutaneous spectrophotometry solutions for hemoglobin measurement are also available but require further evaluation of accuracy, feasibility, and acceptability [50–52]. In addition, the importance of severe anemia as a predictor of active TB [53], as well as the biological mechanism (i.e., hepcidin-driven iron sequestration in the reticuloendothelial system), has been well described [54,55]. Other studies have examined the screening accuracy of C-reactive protein (CRP) compared with WHO 4-symptom TB screen with results differing between studies, settings, and population groups [56–58]. A recent meta-analysis comparing CRP to WHO 4-symptom screen showed comparable sensitivity and higher or similar specificity [59]. The key barrier to use of CRP is that access to POC CRP blood tests is currently very limited [57]. As access to CRP tests expand, future evaluations including CRP in similar risk scores is needed [60].

Measured temperature at >37.5˚C, an additional variable routinely available in LMIC clinic settings, was also independently predictive of TB, indicating the importance of objective measures of fever in addition to patient history [61]. The lack of correlation between reported fever and measured temperature might be due to lack of subjective experience of fever, especially if the fever is chronic or low grade, incomplete understanding of the symptom screening question by the client living with HIV, or patient preference not to acknowledge subjective feelings of fever [44,45].

While the use of electronic medical records (EMRs) are expanding in resource-constrained settings, POC EMR systems that can run regression models retaining continuous rather than categorized variables [62] are not widely accessible in LMIC settings due to logistical and resource constraints of equipment procurements and maintenance, limited and inconsistent access to electricity, and computer literacy capacity [63,64]. Our score that could be used in both paper-based approaches and simple EMR solutions is therefore appropriate for the setting it is designed for [18,39].

A potential advantage of our clinical score over WHO 4-symptom screening tool is that the score could be used by program managers to choose the desired cutoff with associated sensitivity, specificity, NPV, PPV, and NNS. For example, among people starting or restarting ART, among whom mortality risk from undiagnosed disseminated TB remains relatively high, a more sensitive screening tool could help reduce morbidity and mortality [2,65]. Notably, in the XPRES cohort, sensitivity of detecting TB at the initial HIV clinic visit among those who died within 6 months of clinic enrollment increased from 79% with WHO 4-symptom rule to 94% with our clinical score at cutoff ≥2, suggesting the potential for improved early case finding with possible morbidity and mortality reductions [14,16]. However, further evaluation and validation of the screening tool is needed.

In addition, with support from global health donors, many countries are embarking on ambitious TPT scale-up for PLHIV, with the majority of targeted TPT recipients being long-term stable ART patients [9]. Following the 2018 United Nations high-level meeting on TB, the US President's Emergency Plan for AIDS Relief (PEPFAR) committed to reaching >13 million PLHIV with TPT by 2021 [9]. Although increases in NPV by using a clinical score cutoff of ≥2 instead of WHO 4-symptom TB screen are modest, ranging from 0.3% to 1.7%, use of the clinical score cutoff of ≥2 during the proposed TPT scale-up for PLHIV could

potentially avoid 19,500 to 97,500 PLHIV with active TB (assuming 5% TB prevalence) or 39,000 to 195,000 PLHIV (assuming 10% TB prevalence) being inappropriately prescribed TPT. Missed active TB increases morbidity and mortality risk for the patient, but also increases risk of isoniazid-resistant TB [9], which is associated with worse treatment outcomes and may be transmitted to others [10,66]. Our simple screening rule approach to increasing sensitivity and NPV would be much less expensive and logistically challenging than the current WHO recommendation to consider adding CXR, which is not widely available at LMIC clinic settings, to WHO symptom screen [11,67]. In addition, the NPV increase associated with adding CXR to WHO screening rule of 0.9% (at 5.0% TB prevalence) is similar to the NPV increase gained by our much simpler and less costly clinical score at cutoff $\geq 2$ (0.3% to 1.7% at TB prevalence of 4.6% to 18.4%) [6].

Another potential advantage of the clinical score is that the cutoff can be tailored to the use case scenario [68]. As described above, for clients at ART enrollment, reenrollment, or being assessed for TPT eligibility, ruling out active TB is a high priority, and, therefore, high sensitivity and NPV are desired and a cutoff of $\geq 2$ could be chosen. However, for stable patients on long-term ART who have completed a course of TPT, a screening rule cutoff with higher specificity and therefore higher PPV (e.g., cutoff >10) could be chosen to lower the NNS and improve efficiency and cost-effectiveness [18,69,70].

The clinical score could also facilitate differentiated TB care based on TB risk [71]. Firstly, the clinical score is relatively simple and could be used by community healthcare workers in the community [71], with community-based care models for HIV and TB increasingly important to decongest health facilities during the Coronavirus Disease 2019 (COVID-19) pandemic [72]. Secondly, the score could facilitate identifying which new or long-term ART patients should be prioritized for dedicated adherence and retention resources to ensure completion of the TB diagnostic and treatment cascade, with loss to follow-up from HIV–TB care a common problem in LMIC [16,73,74]. Similarly, prioritization of limited on-site GeneXpert diagnostics can be informed by the clinical score to increase cost-effectiveness of POC Xpert use [75]. Finally, diagnostic and therapeutic algorithms could be stratfied by risk groups, with more aggressive TB case finding and treatment approaches appropriate for highest risk groups (e.g., sputum culture, urinary diagnostics, abdominal sonography, or empiric TB treatment) [19]. For example, for patients who are severely ill, and where TB diagnostics are not available or have long turnaround times, a high clinical score of >10, which has a PPV of 22% to 33%, could guide rapid initiation of empiric TB treatment [76].

The dual modeling approach of logistic regression and use of random forest machine learning helps to build confidence in the final practical clinical score for use in LMIC clinic settings. The random forest approach, similar to other machine learning approaches, is better able to capture nonlinear relationships between predictors and outcomes compared with well-established generalized linear regression models [17], because random forest models are not dependent on making assumptions of average linear or curvilinear associations between covariates and outcomes. Among machine learning models, random forest models are particularly strong at predicting categorical outcomes like our TB outcome [77]. For example, despite using the fractional polynomial transformed BMI variable in the logistic regression backward stepwise elimination approach, it was eliminated from the parsimonious model at $p > 0.01$. In contrast, the importance of BMI in discriminating prevalent active TB using the mean decrease in Gini value analysis indicated the importance of BMI in its ability to accurately split groups of patients into those who have or do not have prevalent active TB across the 1,000 decision trees examined in our random forest model. The high ranking of BMI according to mean decrease in Gini value indicates the significant decrease in average, weighted decision tree node purity that occurred when BMI was removed from the possible list of predictor variables [77].

Our analysis also indicates a potential weakness of machine learning approaches. Although the random forest model is superior to single decision trees in reducing the likelihood of over-fitting to the training data [77], we observed extremely high discrimination of the random forest model on the training data and a significant drop in discrimination on the validation data. This highlights the importance of a stringent validation approach [17]. Notably, the most widely available training resources and publications use a random 75%:25% split to create training and validation datasets for random forest models [36], but we purposefully split the dataset into northern and southern clinics in Botswana with a 50%:50% split, which is in line with more stringent validation approaches that help better assess generalizability of predictive models [17,24]. The tendency of the machine learning approach used in this study to overfit to the training data is therefore a limitation of the study. However, this joint approach of using multiple modeling approaches represents a useful contribution to the TB screening literature in line with emerging expert guidance [78,79].

Components of this study that build confidence in the results include the use of data from high-quality prospective cohorts, meaning there was minimal missing covariate data and strong ascertainment of the primary outcome of interest (prevalent active TB) through strengthened ICF in XPRES throughout follow-up and baseline collection of sputum samples for TB diagnosis in all 3 validation cohorts. The study observed high screening accuracy in the 3 external validation cohorts, XPHACTOR, TBFT, and Gugulethu cohorts, representing 3 geographically separate cohorts, with very different cohort characteristics, in very different settings. Limitations include that the risk score has been validated in adult cohorts of PLHIV in sub-Saharan Africa (SSA) and may not be generalizable to pediatric cohorts or cohorts in resource-rich settings like the US and Europe. In addition, our goal was to validate the simple clinical risk score in external validationd datasets, rather than the statistical models; therefore, only the clinical risk scores should be considered externally validated in this analysis. In addition, further evaluation and validation of this approach is needed in SSA, especially in East and West African cohorts. Since only 1 cohort (XPHACTOR) included patients currently on ART, further evaluation in cohorts of long-term ART enrollees would be helpful. Another limitation is that the approaches to TB case finding were different across the 4 cohorts and that for the XPRES and XPHACTOR cohorts a clinical definition of TB was included in the TB outcome definition, whereas for TBFT and Gugulethu cohorts, results of enrollment sputum collection for TB culture and Xpert were used to define the TB outcome. However, model results did not change substantially when we restricted the TB outcome in XPRES and XPHACTOR datasets to microbiologically confirmed TB [18]. In addition, our score does not replace WHO 4-symptom screening rule for active TB, but rather supplements WHO screening rule with additional data variables to help detect asymptomatic (subclinical) TB, allow for flexibility in choosing cutoffs, and allow differentiated care. Additional evaluation is needed to further determine feasibility and added benefit of collecting the additional variables.

Another limitation is that in XPRES, sputum samples for microbiological diagnosis were only obtained from symptomatic XPRES enrollees at enrollment and during follow-up. Although the intensive TB symptom and ICF cascades during repeat visits during 6 months of follow-up in XPRES make it less likely that active TB disease was missed over the course of 6 months of follow-up [26], persistently, subclinical TB disease would have been missed. In addition, in the XPHACTOR dataset, 45% of study participants were excluded from the validation dataset due to missing hemoglobin, and in the TBFT dataset, 29% were excluded because they could not produce sputum for culture at trial enrollment. Comparisons of available patient characteristics between persons excluded versus included in the validation datasets did not indicate notable differences, but additional validation exercises in contemporary cohorts with complete covariates are warranted to further build confidence in the risk score.

In conclusion, following further validation and evaluation steps, this new, simple TB screening clinical score for PLHIV, which is appropriate for both ART-naive and ART-experienced PLHIV in SSA, and which incorporates but does not solely rely on WHO 4-symptom screening rule, is a potential timely addition to practical TB screening approaches in LMIC. The clinical score improves on WHO 4-symptom screening rule's capacity to detect subclinical TB, carrying potential associated morbidity, mortality, and TB transmission reduction benefits [5]. The clinical score provides improved sensitivity and NPV over WHO 4-symptom TB screen, which is needed ahead of intensive global TPT scale-up efforts. Finally, the range of clinical scores allows clinicians and program managers to differentiate patient care and choose cutoffs based on the use case scenario and availability of resources to improve precision and quality of patient-centered care.

## Supporting information

**S1 Fig. XPRES trial profile.**
(TIF)

**S2 Fig. Random forest model variable importance ranking by mean decrease in Gini value in the derivation dataset ($N$ = 2,771).**
(TIF)

**S3 Fig. Comparison of AUROC curves by modeling approach (logistic regression versus random forest), covariate number (15 versus 6 variables), and in derivation versus validation datasets.** AUROC, area under the receiver operating characteristic.
(TIF)

**S4 Fig. Association between temperature modeled as a transformed versus linear variable and prevalent active TB.** TB, tuberculosis.
(TIF)

**S5 Fig. Distribution of clinical risk scores and associated sensitivity, specificity, NPV, and PPV for TB across 4 study cohorts.** NPV, negative predictive value; TB, tuberculosis.
(TIF)

**S6 Fig. Sensitivity, specificity, NPV, and PPV of TB clinical score versus WHO 4-symptom TB screening rule among ART-experienced clinic attendees (XPHACTOR, $n$ = 1612).** ART, antiretroviral therapy; NPV, negative predictive value; PPV, positive predictive value; TB, tuberculosis; WHO, World Health Organization.
(TIF)

**S7 Fig. Sensitivity, specificity, NPV, and PPV of clinical score versus WHO 4-symptom screening rule among those who died within 6 months of clinic enrollment.** NPV, negative predictive value; PPV, positive predictive value; WHO, World Health Organization.
(TIF)

**S8 Fig. Clinical score discrimination according to AUROC curve by study cohort.** AUROC, area under the receiver operating characteristic.
(TIF)

**S1 Table. HIV care clinical follow-up of clients in the Botswana XPRES cohort (2010 to 2015).**
(PDF)

**S2 Table. Comparison of XPRES and external validation datasets.**
(PDF)

**S3 Table. A 15-variable multivariable model in derivation dataset ($N$ = 2,771).**
(PDF)

**S4 Table. Importance of predictors in logistic regression versus random forest modeling approaches.**
(PDF)

**S5 Table. Hosmer–Lemeshow test for calibration of final TB prediction model.** TB, tuberculosis.
(PDF)

**S6 Table. Performance of the TB prediction clinical score in derivation and validation datasets.** TB, tuberculosis.
(PDF)

**S7 Table Sensitivity analyses comparing discrimination between our final risk score and risk scores that (a) were derived from fully standardized bet coefficients, (b) excluded BMI, (c) excluded hemoglobin, and (d) excluded sex and smoking variables BMI, body mass index.**
(PDF)

**S1 Text. XPRES enrollment and follow-up procedures.**
(PDF)

**S1 Protocol. XPRES trial protocol.**
(PDF)

**S1 Checklist. TRIPOD checklist: Prediction model development and validation.** TRIPOD, Transparent Reporting of a multivariable prediction model for Individual Prognosis or Diagnosis.
(PDF)

## Acknowledgments

The authors would like to thank all study participants, participating health facilities, and project team members for their contributions to improving TB and HIV care in Botswana. We also thank Prof. Stephen D. Lawn for his contributions as a mentor, advisor, and subject matter expert prior to his death on September 23, 2016. We also thank Sherri L. Pals for her contributions to the statistical design of XPRES and statistical advice in developing analysis plans.

**Disclaimers:** The findings and conclusions in this report are those of the authors and do not necessarily represent the official position of the US Centers for Disease Control and Prevention.

## Author Contributions

**Conceptualization:** Andrew F. Auld, Tedd V. Ellerbrock, Alison D. Grant, Katherine Fielding.

**Data curation:** Andrew F. Auld, Andrew D. Kerkhoff, Yasmeen Hanifa, Robin Wood, Salome Charalambous, Tefera Agizew, Christopher Serumola, Alison D. Grant, Katherine Fielding.

**Formal analysis:** Andrew F. Auld.

**Funding acquisition:** Andrew F. Auld, Rosanna Boyd, Alyssa Finlay, James C. Shepherd, Tedd V. Ellerbrock, Alison D. Grant.

**Investigation:** Andrew F. Auld, Tefera Agizew, Anikie Mathoma, Rosanna Boyd, Anand Date, Unami Mathebula-Modongo, Heather Alexander, Goabaone Rankgoane-Pono, Pontsho Pono, Alyssa Finlay, James C. Shepherd, Tedd V. Ellerbrock, Alison D. Grant, Katherine Fielding.

**Methodology:** Andrew F. Auld, Yuliang Liu, Ray W. Shiraishi, Christopher Serumola, Tedd V. Ellerbrock, Alison D. Grant, Katherine Fielding.

**Project administration:** Andrew F. Auld, Tefera Agizew, Anikie Mathoma, Rosanna Boyd, Anand Date, George Bicego, Unami Mathebula-Modongo, Heather Alexander, Christopher Serumola, Goabaone Rankgoane-Pono, Pontsho Pono, Alyssa Finlay, James C. Shepherd, Tedd V. Ellerbrock, Alison D. Grant, Katherine Fielding.

**Resources:** Andrew F. Auld, Andrew D. Kerkhoff, Yasmeen Hanifa, Robin Wood, Salome Charalambous, Yuliang Liu, Tefera Agizew, Rosanna Boyd, Goabaone Rankgoane-Pono, Pontsho Pono, Alyssa Finlay, James C. Shepherd, Alison D. Grant, Katherine Fielding.

**Software:** Andrew F. Auld, Yuliang Liu, Ray W. Shiraishi, Christopher Serumola, Alison D. Grant, Katherine Fielding.

**Supervision:** Andrew F. Auld, Tefera Agizew, Anikie Mathoma, Rosanna Boyd, Anand Date, Ray W. Shiraishi, George Bicego, Unami Mathebula-Modongo, Heather Alexander, Goabaone Rankgoane-Pono, Pontsho Pono, Alyssa Finlay, James C. Shepherd, Tedd V. Ellerbrock, Alison D. Grant, Katherine Fielding.

**Validation:** Andrew F. Auld, Andrew D. Kerkhoff, Yasmeen Hanifa, Robin Wood, Salome Charalambous, Yuliang Liu, Tefera Agizew, Anikie Mathoma, Anand Date, George Bicego, Unami Mathebula-Modongo, Heather Alexander, Christopher Serumola, Alison D. Grant, Katherine Fielding.

**Visualization:** Andrew F. Auld, Andrew D. Kerkhoff, Alison D. Grant, Katherine Fielding.

**Writing – original draft:** Andrew F. Auld.

**Writing – review & editing:** Andrew F. Auld, Andrew D. Kerkhoff, Yasmeen Hanifa, Robin Wood, Salome Charalambous, Yuliang Liu, Tefera Agizew, Anikie Mathoma, Rosanna Boyd, Anand Date, Ray W. Shiraishi, George Bicego, Unami Mathebula-Modongo, Heather Alexander, Christopher Serumola, Goabaone Rankgoane-Pono, Pontsho Pono, Alyssa Finlay, James C. Shepherd, Tedd V. Ellerbrock, Alison D. Grant, Katherine Fielding.

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
