## [Editor Report · Decision Letter 0]

17 Dec 2020

Dear Dr Auld, 

Thank you for submitting your manuscript entitled "Risk score for predicting HIV-associated tuberculosis to support case finding and preventive therapy scale-up: A derivation and external validation cohort study using regression and machine learning approaches" for consideration by PLOS Medicine.

Your manuscript has now been evaluated by the PLOS Medicine editorial staff as well as by an academic editor with relevant expertise and I am writing to let you know that we would like to send your submission out for external peer review.

Kind regards,

Thomas J McBride, PhD

Senior Editor

PLOS Medicine

---

## [Decision Letter · Decision Letter 1]

17 Feb 2021

Dear Dr. Auld,

Thank you very much for submitting your manuscript "Risk score for predicting HIV-associated tuberculosis to support case finding and preventive therapy scale-up: A derivation and external validation cohort study using regression and machine learning approaches" (PMEDICINE-D-20-05706R1) for consideration at PLOS Medicine. 

Your paper was evaluated by an academic editor with relevant expertise and sent to independent reviewers, including a statistical reviewer. The reviews are appended at the bottom of this email and any accompanying reviewer attachments can be seen via the link below:

[LINK]

In light of these reviews, we will not be able to accept the manuscript for publication in the journal in its current form, but we would like to invite you to submit a revised version that addresses the reviewers' and editors' comments fully. You will appreciate that we cannot make a decision about publication until we have seen the revised manuscript and your response, and we expect to seek re-review by one or more of the reviewers. 

We hope to receive your revised manuscript by Mar 10 2021 11:59PM. Please email us (plosmedicine@plos.org) if you have any questions or concerns.

Please let me know if you have any questions, and we look forward to receiving your revised manuscript. 

Sincerely,

Richard Turner PhD, for Thomas McBride, PhD

rturner@plos.org

Please remove the name of the lead author from the data statement, noting PLOS' policy on data availability (https://journals.plos.org/plosmedicine/s/data-availability). 

Please remove information on funding from the title page and the end of the main text. In the event of publication, this information will appear in the article metadata, via entries in the submission form. 

Please substitute "sex" for "gender" where appropriate, e.g., at line 21, table 5.

Please add a new final sentence to the "Methods and findings" subsection of your abstract, which should begin "Study limitations include ..." or similar and should quote 2-3 of the study's main limitations. 

After the abstract, please add a new and accessible "Author summary" section in non-identical prose. You may find it helpful to consult one or two recent research papers published in PLOS Medicine to get a sense of the preferred style. 

Please break the TRIPOD checklist out into a separate attached file, labelled "S1_TRIPOD_Checklist" or similar and referred to in the Methods section. 

Please state in the Methods section whether the study had a protocol or prespecified analysis plan, and if so attach the relevant document(s) as a supplementary file or files, referred to in the Methods section. Please highlight analyses that were not prespecified. 

Please avoid claims such as "the first", e.g., at the start of your Discussion section, and where needed add "to our knowledge" or similar. 

More than one paragraph in the Discussion section begins by alluding to "strength(s)" or "advantage(s)" and we suggest avoiding repetition and adopting more measured phrasing. 

We ask you to adapt the paragraph beginning "... some of the weaknesses of machine-learning approaches" to acknowledge more directly that you are summarizing the limitations of the present study. 

In your Discussion section, for example, some phrases suggest that the method is ready for immediate clinical use, e.g., "can be used" and "is ... available for clinicians". Please revisit this wording to avoid promotional language, and it may be appropriate to add a brief explanation of the further evaluation steps needed to ready such an approach for clinical application. 

Generally formatting such as "5 previous studies" should be used, though numerals should not be used at the start of sentences. 

In the reference list, please use journal name abbreviations consistently, e.g., "PLoS ONE" and "Lancet HIV". 

Comments from the reviewers:

*** Reviewer #1: 

This is an interesting study on the prediction of HIV-associated tuberculosis using regression and machine learning approaches. However, there are quite a few major issues needing attention.

1) Development and internal validation of the logistic models are mostly fine, but need to be externally validated which was surprisingly not done. External validation results are most important for a generalisable clinical model and a key indicator for the performance and usability of a prediction model. Authors can refer to the TRIPOD statement for guidance on how to develop and validate clinical prediction models. Also the AUROC needs to be presented with 95% CI.

2) The random forest model is a bit redundant as only offers worse performance in validation compared to the logistic regressions, so not much point to go into the length/details on random forest model. Should focus on the logistic models. Can leave the random forest part out or briefly mention it and include the results in the supplementary information.

3) The whole clinical risk score part is confusing, not justified and basically redundant. Firstly, in the current time, logistic prediction models can be easily developed in an app and then being used straightforward in mobile phones and PCs, therefore not much point on the traditional risk scores on dichotomisation of variables which normally offers inferior performance in prediction accuracy as only use partial information (categorical vs continuous data); Secondly, the use of clinical scores for prediction is not justified especially statistically. Do they offer better AUROC or calibration as compared to logistic models in the external validations sets? Thirdly, the presentation of sensitivity, specificity and etc with different cut-offs is all over the place, looks messy and difficult to follow.

4) The abstract could be more concise and focused on the main findings.

5) Title could also be shorter and focused on the main message.

*** Reviewer #2: 

In this paper, the authors have developed a screening tool that can be applied to both HIV positive and negative patients to determine their risk of active TB. This appears to be a very useful study and would make a great contribution to the literature. I have some concerns that I have outlined below.

Major/general comments:

1. The paper is very dense, which makes it hard to read and hard to pull out the most important pieces. I found it hard to keep the characteristics of the different populations in the three external validation sets in my head as I read through the manuscript and so on seeing sensitivity etc results in the text and in figures I felt I was reading without knowing what I would expect, especially relative to each other - e.g. which ones had people on ART? Etc. I don't know quite how to handle this. Maybe an easy look up table showing important differences between the groups - something less busy than Table 1 (and supp table 3)? Maybe a little more help in the text with interpretation?

2. It sounds a little confusing when it says "when the clinical score replaced the WHO four-symptom TB screen" and where the clinical score is compared to the WHO screen as though they are entirely separate. Isn't the WHO screen included in the clinical score? Doesn't this imply that instead of replacing the WHO screen, the clinical score augments it? Therefore, it would make sense that the clinical score is unlikely to be worse than the WHO one. The question is more about how much better it is and whether it warrants the collection of additional information for the improvement it provides. I think it is worth it but I think the wording could be clearer in the comparisons that this is not an entirely separate score.

Specific/minor comments:

Abstracts:

1. It would be helpful if the author could be more clear about the aim of developing the screening tool in the background. It currently states that screening tools are needed to facilitate prudent scale-up of TPT but the proposed screening tool does not appear to identify people with TB infection (rather than TB disease). Do the authors mean that by helping to diagnose more people with TB disease, this allows health workers to avoid incorrectly giving those patients TPT? Thus, avoiding risk of monotherapy and possible resistance?

2. It's not clear what the points in the abstract are (methods and findings section, next to the predictors). Does this mean that when using the score, one would give a person 6 points if they had a temperature over 37.5C, for example? If so, does that mean that just having a temperature and no other indicators would immediately put someone into the >2 clinical score group?

Introduction: No comments

Methods:

1. It's unclear why temperature would be included given that it is already in the WHO four-symptom score. Why was it included separately? Was it collinear to some degree with the WHO score in the regression? And do the authors know why it came out as significant despite the WHO score being in the model?

2. (line 230) A cut-off of P<0.01 was used to choose variables for the model. Did the authors try 0.05 to see what difference it made?

3. (line 232) The authors say that a forward stepwise approach was also tried but I could not find where in the manuscript or supplement these results were. Were they different from the backward approach?

4. (line 233) The authors state that when two variables were highly correlated, one was chosen for inclusion in the model. How did they choose?

5. On line 239, the authors state that the used the bestmtry function. I know at the end of the section the software applications used are listed but I think it's worth stating here which application this function was from.

6. Line 268 states how the clinical score was generated and they say it was from the logistic regression by rescaling the coefficients. Were the standard errors of the coefficients taken into account at all? i.e. to give the better estimated coefficients more weight? And how was the random forest analysis used? It sounds as though, if it brought up a variable that did not come through in the logistic regression, this variable was added into the regression regardless of p-value - is this correct?

7. It would be helpful if the authors could explain what gini is

Results:

1. In table 1, two of the external cohorts are described as having 0 people on ART at time of study enrolment. Were any put on ART at that point, or soon after? If so, did that impact whether or not they developed and were diagnosed with TB in the following 6 months?

2. Figure 6. It was unclear what "screen in" meant.

3. Figure 8. Were there sensitivity analyses using different categories for the score (i.e. cut-offs points other than 2 and 10)? What difference did different cut points make?

Discussion:

1. The first paragraph refers to flexibility but is there a guide for users? To know how to incorporate flexibility and get what they need for their setting?

2. The end of the first paragraph talks about the benefit of incorporating machine learning but my understanding is that machine learning only added one variable to the score (BMI) - is this correct? If so, how well did the score perform without BMI? i.e. how useful was it ultimately to use ML here?

3. The second paragraph refers to the blood test needed. Since this seems marginally harder to obtain than the other information, was there a sensitivity analysis to see how well the clinical score performed with the haemoglobin concentration variable?

4. Further down the discussion, the following sentence: "Although increases in NPV by using a clinical score cut-off of ≥2 instead of the WHO four-symptom TB screen are modest, ranging from 0.3% to 1.7%, use of the clinical score cut-off of ≥2 during the proposed TPT scale-up for PLHIV could potentially avoid 19,500-97,500 PLHIV with active TB (assuming 5% TB prevalence), or 39,000-195,000 PLHIV (assuming 10% TB prevalence) being prescribed TPT." 

I think it is worth adding in "incorrectly" (or "inappropriately") between "being" and "prescribed" at the end of this sentence to be completely clear. Are these estimates *in addition* to what would be achieved with just the WHO score?

5. The sentence following the one quoted above refers to the benefits assuming that IPT would always be used for these patients. Are no other TPT regimens used in these settings?

6. Does this tool only apply to adults? If so, I think this warrants a comment in the discussion.

*** Reviewer #3: 

Thank you for the opportunity to review this fascinating paper on the derivation and validation of a clinical risk score to predict HIV-associated tuberculosis, using secondary study data from Botswana and South Africa. Regression and machine learning approaches were used to derive the score. A few comments should be addressed before publication. 

Comments

From the supplementary text (table 3), it seems as if only symptomatic patients were tested for TB in the XPRES trial. If this is indeed the case, none of these patients would have been subclinical and all would have screened positive using the WHO screening algorithm. This is in contrast with the other three studies, where all participants were tested, regardless of symptoms, and would mean the four study populations would be considerably different. Please clarify. 

Following on the previous point, could the authors clarify how the score would improve subclinical TB case finding? Is it purely because of the inclusion of other clinical characteristics in addition to symptoms? It was not shown in the analysis that subclinical TB would be identified earlier/better, especially because the main outcome included microbiological and clinical TB? 

For the derivation of the score, please clarify what the importance of the mean decrease in accuracy was from the random forest model. The text compares the mean decrease in Gini to the logistic regression model, but does not mention the mean decrease in accuracy and it seems quite different from the Gini values. 

When looking at the derivation of the score and the six final variables which were included, it seems from the supplementary appendix 5 that two variables, prior TB and respiratory rate, might have been good candidates for inclusion as well. Could the authors, by using this table, explain how the final six variables were selected? 

Was there a specific rationale why only the number of WHO symptoms, and not specific individual symptoms or combinations of symptoms, was tested in the models? 

Could the authors speculate on the possible inclusion of a point-of-care CRP test? 

Did the authors consider the imputation of missing values for the validation datasets? 

How was the risk score cut-offs determined? Could this be indicated clearly in Figure 9; at the moment it is not clear if the NNS refer to 2, 2-10, and 11, or <2, 2-10, and >10. 

Would it be possible to include 2-3 sentences in the discussion on the practical application of the risk score: for instance, does it need to be tested in an implementation science study before roll-out or are the authors convinced of its value in a programmatic setting, taking into account that the derivation and validation datasets were from trials and cohorts? 

Please clarify whether ethics approval was given for this specific analysis, or whether it was not deemed necessary since additional analyses were included in the original informed consent documents. 

Discretionary remarks

Please include a flow diagram of the XPRES trial in the supplementary text to make it easier for the reader to understand. 

Supplementary table 8: correct the total number of participants for XPHACTOR (1807 or 1811?)

Line 209: weight was included in the tables. 

Could the type of variable transformations be specified, if a reader would like to replicate the work? 

*** Reviewer #4: 

The manuscript is well-written and clearly presented overall. The proposed clinical score may be useful for TB screening in people living with HIV. However, two variables (Demographics-Education-higher and HIV-TB history-Previous TB treatment) in Table 2 have a p value of less than 0.001 in univariable regression analysis but were not selected for multivariable logistic regression analysis. Authors need to explain this reasonably.

*** Reviewer #5: 

This is an interesting and well-written paper on developing a risk score for tuberculosis in people living with HIV. Overall, the subject is important, the methodology sound (including comparison of several methods and variable selection approaches), and the performance of the score has been tested in external validation. I particularly like about this paper that it compares conventional logistic regression with machine-learning approaches (eventually, the latter do not outperform the former but nevertheless give some guidance regarding the inclusion of prediction variables). I have three concerns and suggestions:

1) The newly developed score does not seem to clearly outperform the the simpler WHO four-symptom screen (according to the external validation shown Figure 6 and 7); especially it never exhibits simultaneously (clearly) superior sensitivity and specificity . If my reading of the results is correct, then it is unclear how much use this score will find in practice. The authors argue in the discussion that the score offers the advantage (compared to the WHO screen) of being more flexible and allowing to adapt the level of specificity and sensitivity to the specific setting (by considering more liberal or more conservative thresholds). I was wondering in this context to what extent this is also possible within the WHO screen by changing the number of symptoms required for a "positive"? It is unclear however whether this benefit will justify the additional complexity/effort associated with the newly developed score. 

2) The authors refer to a 15-covariate multivariable logistic regression model. It would be good to add this to table 2. 

3) Some differences between the "final adjusted regression" model in table 2 and the multivariable model in table 3 are a bit surprising and would be worth assessing or discussing in more detail (some of those differences might be due to the binarisation of some variables; but I am not sure that this explains all differences). For example sex and smoking are highly significant in the model shown in table 1 but not or only borderline significant in table 3. Why? Based on this I was also wondering whether it would be worth simplifying the score and remove those two factors?

***

[LINK]

---

## [Decision Letter · Decision Letter 2]

7 Jul 2021

Dear Dr. Auld,

Thank you very much for re-submitting your manuscript "Risk score for predicting HIV-associated tuberculosis to support case finding and preventive therapy scale-up: A derivation and external validation cohort study using regression and machine learning approaches" (PMEDICINE-D-20-05706R2) for consideration at PLOS Medicine.

I have discussed the paper with our academic editor and it was also seen again by three reviewers. I am pleased to tell you that, provided the remaining editorial and production issues are dealt with, we expect to be able to accept the paper for publication in the journal.

[LINK]

Please let me know if you have any questions, and we look forward to receiving the revised manuscript.   

Sincerely,

Richard Turner, PhD

rturner@plos.org

Requests from Editors:

We suggest some restructuring and trimming of the title: "Derivation and external validation of a risk score for predicting HIV-associated tuberculosis to support case finding and preventive therapy scale-up: A cohort study".

Early in the abstract, please remove "... to over 13 million PLHIV by 2021" as the context for this target appears elsewhere.

In the abstract, please briefly mention when the XPRES data were collected. 

Please also quote aggregate demographic details for XPRES participants.

Noting one instance in figure 5, please substitute "sex" for "gender" throughout the ms, where appropriate.

Noting the comments from one referee, we ask you to convert move several of the figures to supplementary information: we suggest figures 2, 4, 6, 7 and 8.

Please adapt the attached TRIPOD checklist so that individual items are referred to by section (e.g., "Methods") and paragraph number, not by page or line numbers as these generally change in the event of publication. 

Comment from academic editor:

I think that the authors should cite and mention an article published last year, listed below, which belongs with the list of other TB risk prediction tools in the Discussion.

"Baik Y, Rickman HM, Hanrahan CF, Mmolawa L, Kitonsa PJ, Sewelana T, et al. (2020) A clinical score for identifying active tuberculosis while awaiting microbiological results: Development and validation of a multivariable prediction model in sub-Saharan Africa. PLoS Med 17(11): e1003420. https://doi.org/10.1371/journal. pmed.1003420."

Comments from Reviewers:

*** Reviewer #1: 

Thanks authors for their great effort to improve the manucript. I am mostly satisfied with the response and revision so no further major issues needing attention. However, although the authors argued strongly I still reserve my opinion to make the paper more concise and readable by especially making the abstract and title more concise and focused on main messages (shorter) so relatively easy to follow. But it's up to the editors to decide and not a major issue.

*** Reviewer #3: 

Thank you for the revision, no additional concerns.

*** Reviewer #5: 

[supportive report received]

***

[LINK]

---

## [Editor Report · Decision Letter 3]

21 Jul 2021

Dear Dr Auld, 

On behalf of my colleagues and the Academic Editor, Dr Rosen, I am pleased to inform you that we have agreed to publish your manuscript "Derivation and external validation of a risk score for predicting HIV-associated tuberculosis to support case finding and preventive therapy scale-up: A cohort study" (PMEDICINE-D-20-05706R3) in PLOS Medicine.

Prior to final acceptance, we suggest making that "... median age was 34 years" (abstract); and at various points in the ms we wonder whether "decrease in Gini" could become "... decrease in Gini score", "... level" or similar?

PRESS

Sincerely, 

Richard Turner, PhD 

rturner@plos.org